# Enhanced Extracellular Transfer of HLA-DQ Activates CD3^+^ Lymphocytes towards Compromised Treg Induction in Celiac Disease

**DOI:** 10.3390/ijms23116102

**Published:** 2022-05-29

**Authors:** Michael Hudec, Iva Juříčková, Kamila Riegerová, Saak V. Ovsepian, Marie Černá, Valerie Bríd O’Leary

**Affiliations:** 1Department of Medical Genetics, Third Faculty of Medicine, Charles University, Ruská 87, 10000 Prague, Czech Republic; iva.jurickova@lf3.cuni.cz (I.J.); marie.cerna@lf3.cuni.cz (M.Č.); valerie.oleary@lf3.cuni.cz (V.B.O.); 2Department of Immunology, Third Faculty of Medicine, Charles University, Ruská 87, 10000 Prague, Czech Republic; kamila.riegerova@lf3.cuni.cz; 3Faculty of Engineering and Science, University of Greenwich London, Chatham Maritime, Kent ME4 4TB, UK; s.v.ovsepian@greenwich.ac.uk

**Keywords:** exosome, major histocompatibility complex II, autoimmunity, monocyte-derived dendritic cells, gluten

## Abstract

Celiac disease (CeD) manifests with autoimmune intestinal inflammation from gluten and genetic predisposition linked to human leukocyte antigen class-II (HLA-II) gene variants. Antigen-presenting cells facilitate gluten exposition through the interaction of their surface major histocompatibility complex (MHC) with the T cell receptor (TCR) on T lymphocytes. This fundamental mechanism of adaptive immunity has broadened upon recognition of extracellular exosomal MHC, raising awareness of an alternative means for antigen presentation. This study demonstrates that conditioned growth media (CGM) previously exposed to monocyte-derived dendritic cells from CeD significantly downregulates the CD3^+^ lineage marker of control T cells. Such increased activation was reflected in their elevated IL-2 secretion. Exosome localization motif identification and quantification within *HLA-DQA1* and *HLA-DQB1* transcripts highlighted their significant prevalence within *HLA-DQB1* alleles associated with CeD susceptibility. Flow cytometry revealed the strong correlation between HLA-DQ and the CD63 exosomal marker in T cells exposed to CGM from MoDCs sourced from CeD patients. This resulted in lower concentrations of CD25^+^ CD127^−^ T cells, suggestive of their compromised induction to T-regulatory cells associated with CeD homeostasis. This foremost comparative study deciphered the genomic basis and extracellular exosomal effects of HLA transfer on T lymphocytes in the context of CeD, offering greater insight into this auto-immune disease.

## 1. Introduction

Celiac disease (CeD) is a chronic autoimmune disorder of the small intestine with etiological symptoms manifesting as fatigue, weight loss, anemia, nausea, constipation or diarrhea [1]. Pathology includes villous atrophy, crypt hyperplasia, inflammatory cell infiltration and activation [2]. CeD develops upon the ingestion of gluten, a structural wheat protein also found in barley, rye and oats [3]. Gluten is incompletely cleaved into short peptides by gastric, pancreatic and brush border enzymes that are presented in proximity to the lamina propria of the small intestine, eliciting the basis of the CeD auto-immune response [4,5,6,7]. Affecting 1–2% of Caucasian populations, CeD is prompted by familial inheritance predisposition to auto-immunity [8] and selective genotypic status involving human leukocyte antigen (HLA) class-II DQ2 (DQA1 *05:01/DQB1 *02:01) and DQ8 (DQA1 *03:01/DQB1 *03:02) alleles [9,10,11,12].

The HLA class-II (HLA-II) heterodimeric surface molecule or major histocompatibility complex (MHC) is expressed on macrophages, B-lymphocytes and monocyte-derived dendritic cells (MoDCs), qualifying their professional antigen-presenting cell (APC) status [13]. In CeD, gluten peptides are presented by APCs through HLA-DQ, enabling the direct interaction with the T cell receptor (TCR) of CD4^+^ T lymphocytes, eliciting a proinflammatory cytokine-rich microenvironment [14,15,16]. This conventional mechanism of direct interaction between APC and T cells via the MHC-TCR bridge in healthy individuals has evolved with the recognition of a potential role for extracellular exosomes in this process, especially in the context of CeD autoimmunity. Exosomes hold a variety of cargo, but, importantly, they can also present antigens by harboring HLA class-II molecules enabling the activation of CD4^+^ T lymphocytes [17,18,19]. MoDCs release extracellular exosomes [18,20,21,22], instigating an immuno-inhibitory or -stimulatory function, contingent with donor status [23].

A subpopulation of T-regulatory cells (Tregs) originates from conventional CD4^+^ T lymphocytes, following induction by antigen recognition and IL-2 cytokine release [24]. Maintaining tissue homeostasis, Tregs are important for the initiation and maintenance of peripheral tolerance in the prevention of excessive immune responses, such as that observed with auto-immunity [25]. While the direct intercellular response of APC and T cells via MHC-TCR has been well characterized [26], the response of Tregs to extracellular exosomes within the context of tolerance loss to gluten remains to be elucidated.

This study serves to decipher the effects of conditioned growth media (CGM) from CeD sourced MoDCs in comparison to healthy controls (CTLs) on T cells. *In silico* analysis reveals the genomic priming of *HLA-DQ* alleles associated with CeD risk for exosomal transport. Despite the over-activation of T cells by CeD-sourced exosomes, our findings nevertheless reveal fundamental limitations in Treg transformation capability, shedding light on the basis of this auto-immune condition of the gut.

## 2. Results

### 2.1. T cell Activation upon Direct Contact with MoDCs of CeD Patients Is Circumvented by Exposure to CGM

MHC-TCR engagement binds the MoDC to the T -lymphocyte in the act of antigen presentation. This hallmark mechanism of the adaptive immunity has been exploited through the cultivation and differentiation of CeD and CTL monocytes into MoDCs, as previously reported [10]. The co-cultivation of such MoDCs with T cells enabled the monitoring of viability and intercellular interaction (Figure 1A). Limited proximity within 24 h co-cultivation was evident in contrast to a 2- fold increase in the number of MoDC—T cell contacts recorded (1 ± 0.1 versus 3 ± 0.5 contacts, respectively) between days 3 and 6.

The potential for T cell activation in the absence of MoDCs was determined using CGM derived from the in vitro cultivation of CeD- or CTL-sourced MoDCs, in comparison to naïve growth media with/without LPS. (Figure 1B left). Intriguingly, by cultivation day 6 in cell-free CGM, T cells diversified into two populations represented by CD3^+^ upregulated (green; non-activated) or downregulated (orange; activated) expressions (Figure 1B). While T cells were activated by complete media with the inclusion of the human granulocyte-macrophage colony-stimulating factor (CMI; 24 ± 2%), in contrast, a significant 1.7-fold increase in the percentage of CD3^+^ T cells was noted following their exposure to CGM derived from CTL—MoDCs (74 ± 2%, *p* = 0.05; Figure 1C). This significant difference was even more evident with 90 ± 1% of T cells in an activation state when cultivated for 6 days in CGM derived from CeD MoDCs (*p* = 0.0003; Figure 1C). It should be noted that competitive binding between HLA-DQ and anti-CD3 may have influenced the result showing the predominance of the CD3 downregulated T cell population. Nevertheless, these finding were supported by their increased IL-2 secretory profile in comparison to non-conditioned stimulated media (i.e., LPS containing CMI; *p* = 0.004) or CTL-CGM (*p* = 0.05) and provided further evidence of their activation (Figure 1D). These results highlight that, while T cells could interact with MoDC, direct contact between these cells of the adaptive immunity is not necessary for T cell stimulation, but rather they can also be activated via indirect extracellular means. Furthermore, CGM derived from CeD patients held an increased capacity for T cell activation and cytokine secretion.

### 2.2. Higher Prevalence of Exosome Localization Motifs (ELMs) in HLA-DQA1 and HLA-DQB1 Alleles Associated with CeD Risk

Extracellular vesicles from monocyte-derived dendritic cells are enriched in exosomes [27] and have been shown to carry functionally active molecules of HLA class I/II complexed with antigens [28]. *HLA-DQA1* and *HLA-DQB1* are located on chromosome 6p21.32 with their respective transcription running in opposite directions within this genomic region (Figure 2A). While both genes contain up to five exons, *HLA-DQA1* is larger than *HLA-DQB1* by 10,250 bps (Figure 2A). ELMs have been reported within non-coding RNA destined for extracellular transport [29,30]. Given such a background, this study conducted an in-depth investigation of both *HLA-DQA1* and *HLA-DQB1* loci for the previously reported ELMs [29,30] to ascertain their prevalence and association with CeD-risk alleles. IPD-MHC database sequences enabled the retrieval of *HLA-DQA1* and *HLA-DQB1* protection/risk transcript alleles [31] associated with CeD from exons 1–5 to be scanned for motifs (1—ACCAGCCU; 2—CAGUGAGC; 3—UAAUCCCA). While evidence was found for the single presence of the entire 8 bp ACCAGCCU motif within exon 3 (encoding the HLA extracellular domain) of *HLA-DQA1*01:02* and *HLA-DQA1*01:03*, shorter ≥5 bp versions were more evident within this region for these and other *HLA-DQA1* alleles (Figure 2B). These ELMs showed high prevalence within the *HLA-DQA1* exon 4 transcript encoding the transmembrane cytoplasmic tail domain in contrast to that found in *HLA-DQB1*, where they were not found to the same extent. Rather, the *HLA-DQB1* leader sequence showed a greater prevalence of truncated CAGUGAGC in comparison to *HLA-DQA1*. Quantitative data for these truncated ELMs within all protective or risk alleles were combined. The ELMs ACCAG, ACCAGC, ACCAGCC were found to be significantly (*p* = 0.03) more prevalent in *HLA-DQB1* alleles associated with CeD risk in comparison to those linked to protection against this condition (Figure 2C). Likewise, CAGUG, CAGUGA, CAGUGAG were also significantly (*p* = 0.02) predominant within *HLA-DQB1* alleles associated with CeD risk in contrast to *HLA-DQA1* (Figure 2C). To the best of our knowledge, this is the first report of ELM identification within the hallmark *HLA-DQA1* and *HLA-DQB1* of the adaptive immunity. In addition, these findings highlight the significant higher frequency of ELMs within *HLA-DQB1* alleles associated with CeD susceptibility.

### 2.3. Enhanced Exosomal HLA-DQ in T cells Exposed to CGM from CeD-Sourced MoDCs

It has been reported that specific *HLA* alleles predict T cell activation [32]. Given the greater stimulation of CTL T cells to CGM from CeD-MoDCs reported above and the increased risk for CeD associated with exosomal motif prevalence in specific *HLA* alleles, an examination of exosomal HLA-DQ in activated T cells was conducted. The minor percentage of HLA-DQ associated with exosomes in T cells exposed to CMII (i.e., CMI with, instead, exosome-depleted fetal bovine serum; 1.2 ± 0.3%) can perhaps be explained by a negligible amount of FBS still evident in the GM (Figure 3A, left). Nevertheless, HLA-DQ correlated with the CD63 exosomal marker in 37.4 ± 2% of the CTL T cell population as observed from the scatter plot analysis (Figure 3A, middle) and Western blot detection (Appendix A). Confocal micrographs provided evidence for the presence of HLA-DQ within such T cells, in comparison to the naïve experimental group (Figure 3A). A highly significant 0.9-fold increase (71.3 ± 4%; *p* = 0.0002; *n* = 3) was evident in similar T cells post exposure to CGM from CeD-MoDCs (Figure 3A, right). HLA-DQ was also detected by Western blotting of exosomal fractions (Appendix A). Likewise, confocal microscopy provided further evidence of an enhanced presence of HLA-DQ in these T cells, compared to the other experimental groups localized to the cytoplasm rather than the nucleus (Figure 3A). Such findings provide evidence that CeD may potentially operate an enhanced HLA-DQ exosomal transportation mechanism compared to healthy CTLs, presumably for antigen presentation to T cells in addition to conventional MHC–TCR complex interaction.

### 2.4. Compromised T cells Induction to Tregs following Exposure to CGM from CeD-Sourced MoDCs

Regulatory T cells (Tregs) are induced in the periphery from naïve T cells, which assist the suppression of immune responses, enabling homeostasis and self-tolerance [33]. Decreased frequency of this Treg sub-lineage T cell population may therefore lead to the development of auto-immune disease [34]. It has been reported that in vitro-induced Tregs can also suppress auto-immune conditions in vivo [35]. Given the differential status of T cell activation after exposure to CGM from CeD compared to CTLs shown above, this study sought to determine the potential Treg presence within these T cell populations using conventional CD25 and CD127 markers [36] (Figure 4). A comparison of the percentage cell counts using area under the curve analysis of curved flow cytometric histograms (CFCHs) indicated a significant 6.26% (2591/41,327 arbitrary units (AU), *p* = 0.029) difference between T cell populations exposed to CTL-CGM compared to CeD-CGM (Figure 4A,B), indicative of the presence of Tregs with expected upregulated CD25 expression (Figure 4B). Although mean fluorescence intensity (MFI) values for CD25 were higher in T cells exposed to CeD-CGM compared to naïve T cells exposed to NGM (*p* = 0.03), the highest values were found following exposure to CTL-CGM (*p* = 0.01) (Figure 4C). The downregulation of CD127 was also more evident within T cells exposed to CTL-CGM, highlighting the presence of Tregs within this population (Figure 4D). Area under the curve analysis for CD127 revealed a 4.6% (1737/37,617 AU, *p* = 0.02) differential when compared to T cells exposed to CeD-CGM (Figure 4D). Mean fluorescence intensity results revealed significantly reduced CD127 levels in T cells exposed to CTL-CGM in comparison with T cells exposed to CeD-CGM (*p* = 0.03; Figure 4E). Non-significant differences were found between mean fluorescence intensity levels for CD127 when T cells exposed to CeD-CGM and NGM were compared (Figure 4E). Despite the limitation of the small sample size, findings still identified Tregs subsets within a T cell population exposed to CGM sourced from healthy individuals. Importantly, the absence of Treg markers within a T cell population exposed to CGM sourced from CeD patients is perhaps indicative of compromised T-reg induction or inhibited Treg-marker expression due to inhibitory secretory factors originally emanating from CeD-MoDCs.

## 3. Discussion

Conventional cell-mediated immune responses eliminate antigen fragments by direct contact between APCs and effector T cells [37]. While this study recognized increased contact between MoDCs and T cells with time in co-culture, T cell activation occurred solely with cell-free CGM. Importantly, this investigation noted that CGM sourced from CeD patient MoDCs (CeD-CGM), in comparison with CTL-CGM, elicited alternative responses in T cells derived from healthy individuals. It also identified the enhanced uptake of HLA-DQ within such T cells stemming from CeD-CGM exosomes. In silico analysis explored the genomic basis for preferential extracellular processing of HLA-DQ within *HLA-DQA1* and *HLA-DQB1* alleles associated with CeD risk, and recognized the significant increased prevalence of exosomal localization motifs. Despite increased stimulation and IL-2 secretion, healthy control T cells exposed to CeD-CGM exosomes elicited compromised induction towards Treg transformation pointing towards an extracellular basis for transportable auto-immunity.

CD86 upregulation in CeD-MoDCs has been previously reported [10]. This potent T cell co-stimulatory molecule, along with HLA class-I and -II complexes, have been found in MoDC exosomes within preformed microdomains, prompting an examination of their immunostimulatory capacities [22]. The source of such exosomes has been debated with the speculation that they arise from endocytic compartments or the external milieu to assist immune synapse formation [22,38]. Adding to this complexity, this investigation revealed that exosomes secreted from CeD-MoDCs within cell-free CGM elicited alternative responses towards T cells compared to CTL-MoDCs, offering an insight into the basis for immune dysregulation associated with this condition. Instigating increased CD3^+^ T cell activation and IL-2 secretion, CGM from CeD-MoDC contained a higher proportion of exosomes loaded with HLA-DQ than CGM from CTL-MoDC.

Collectively, 40% of the genetic risk for CeD development is conferred by *HLA* genes [39]. More specifically, CeD is strongly associated with *HLA-DQ* [40], particularly in patients from the Czech Republic [10]. Given that monocytes were isolated from the blood of this CeD patient population and the recognized role of exosomes as MHC/HLA vehicles, this foremost study deciphered *HLA-DQA1* and *HLA-DQB1* for exosome targeting/localization sequences. Nucleotide *cis*-acting elements or motifs that prime transcripts for exosome packaging have been identified in long non-coding RNA [29,30]. While in silico findings revealed in this investigation that truncated ELMs exist in *HLA* alleles linked to protection, a significant over-representation of ELMs was identified within *HLA* alleles associated with CeD risk. The manipulation of ELMs within *HLA* genes will form the basis of future research to decipher the immunological implications of this foremost finding on the reprogramming of target cells.

It has been reported that the majority of MHC class II containing exosomes on the T cell surface neither fuse with the plasma membrane nor are efficiently internalized [20]. This view evolved with the recognition that DC exosomes can be efficiently transferred to activated T cells, irrespective of their TCR specificity, and rather recognized leukocyte function-associated antigen-1 (LFA-1) as a determinator of exosome acquisition efficacy by recipient T cells [21]. Our data show the increased presence of HLA-DQ in T cells after exposure to CeD-CGM, in comparison to CTL-CGM or NCM. While almost no HLA-DQ was evident in naïve T cells exposed to NCM, the signal was concentrated on the plasma membrane in T cells exposed to CTL-CGM. Intriguingly, the intercellular distribution of HLA-DQ was located throughout the cytosol rather than solely on the plasma membrane, potentially indicating exosomal confinement rather than membrane fusion in the case of CeD-sourced CGM.

Naïve CD4^+^ T cells at the periphery can differentiate into induced Tregs that constitutively express CD25^+^ (interleukin-2 receptor) and play a central role in preventing the activation of autoreactive T cells [41]. While representing only a small percentage of Tregs as a whole, this subset is highly enriched in the gut and is particularly important for tolerance against food allergens and helps maintain immune homeostasis during environmental alteration [42,43]. Importantly, their absence exacerbates pathogenesis in several models of mucosal autoimmunity, such as CeD [44]. Upregulated CD25 expression became evident in T cells only exposed to CTL-CGM indicative of their induction towards Tregs under these conditions. The notable absence of this effect in T cells exposed to CeD-CGM or NGM was potentially indicative of a compromised or lower stimulation status towards Treg induction. CD127 expression inversely correlates with the human Treg autoimmune suppressive function [45]. The results indicate that this Treg CD127 marker is evident in T cells only exposed to CTL-CGM and not CeD-CGM. It can be speculated that exosomes or another secretory factor within CGM from CeD patients inhibits Treg induction.

While understanding tissue-resident Treg cells in the development and maintenance of human autoimmunity is important, this investigation highlighted the extracellular role of HLA as determined by exosome localization nucleotide sequences to influence peripheral Treg induction and offers a wider perspective on the concept of genetic CeD susceptibility. The manipulation of genetic zip-codes responsible for exosome delivery may divert the consequences of *HLA*-risk allele expression assisting the restoration of immunological tolerance to gluten and represent the elusive cure for CeD association linked to familial autoimmune inheritance.

## 4. Materials and Methods

### 4.1. Clinical Diagnosis of CeD Patients and Pedigree Overview

CeD was diagnosed in this study population following the current guidelines, i.e., measurement of total serum IgA antibodies against tissue transglutaminase 2 (tTGA-IgA) being 2–10 times the upper limit of normal (10 × ULN) and biopsy examination [46]. Written informed consent was obtained from all individuals involved in this study with approval from the Ethics Committee of the Institute for Clinical and Experimental Medicine and Thomayer Hospital, Prague, Czech Republic (approval number: G-18–23). A total of 11 participants (*n* = 11) were enrolled, as previously reported [10], and categorized as follows: Gluten-free diet (GFD)-treated CeD patients (CeD: *n* = 4); normal controls with no known auto-immune disease, inflammation or malignancy (CTL: *n* = 7). All subjects were Caucasians of European ancestry.

### 4.2. Monocyte Isolation from Peripheral Blood Mononuclear Cells

This procedure follows what was previously reported [10]. Peripheral whole blood was collected from CeD and CTL participants (outlined above) by a phlebotomist employed in the Department of Medical Genetics of the Third Faculty of Medicine, Charles University, Prague, Czech Republic. Following dilution with RPM1 1640 medium (1:1; Sigma-Aldrich, St. Louis, MO, USA), whole blood was gently layered over Ficoll–Paque in a 1:0.7 ratio (GE HealthCare Bio-Sciences AB, Uppsala, Sweden) in a Falcon tube. Following centrifugation for 30 min at 400× *g*, distinct layers were formed. The presence of peripheral-blood mononuclear cells (PBMCs) at the interface of the top and second layers beneath the plasma zone were noted. Using a pipette, the plasma layer was removed, and PBMCs transferred to room temperature (RT)-complete RPM1 1640 medium (Sigma-Aldrich, St. Louis, MO, USA) to remove any remaining platelets. Following centrifugation for 20 min at 400× *g*, samples of PBMCs were stained with trypan blue, viewed under a light microscope and counted on a hemocytometer slide. PBMCs (10^6^ cells) were cultivated for 2 h in a 75 cm^2^ plastic culture flask (Schoeller Pharma, Prague, Czech Republic) in complete medium (CM) containing RPMI 1640, fetal bovine serum (10%; Sigma-Aldrich, St. Louis, MO, USA) in the presence of antibiotic–antimycotic solution (1%; Sigma-Aldrich, St. Louis, MO, USA) at 37 °C in a 5% CO_2_ atmosphere and 95% relative humidity. The non-adherent fraction was then removed following washing with CM.

### 4.3. Generation of Monocyte-Derived Dendritic Cells (MoDCs) and Conditioned Growth Media (CGM)

Adherent monocytes were cultivated in CMI (CM with inclusion of human granulocyte-macrophage colony-stimulating factor (GM-CSF) (500 ng/mL; Sigma-Aldrich, MO, USA) and human interleukin-4 (IL-4) (200 ng/mL; Sigma-Aldrich, St. Louis, MO, USA)) for 6 days at 37 °C in 5% CO_2_ and 95% relative humidity in a 75 cm^2^ plastic culture flask (Schoeller Pharma, Prague, Czech Republic), as previously reported [10]. CMI was replaced with CMII (i.e., CMI with exosome depleted fetal bovine serum instead (Thermo Scientific, Waltham, MA, USA, cat. A2720801)) on day 3. Generated MoDCs (CD11c^+^, CD14^−^) were divided into 6-well plates (Schoeller Pharma, Prague, Czech Republic) with the addition of lipopolysaccharide (LPS) (100 ng/mL; Sigma-Aldrich, St. Louis, MO, USA) and cultivated for 24 h to induce MoDC maturation. Mature MoDCs were harvested by a cell scraper and briefly centrifuged at 400× *g*. Harvesting of CGM then occurred, i.e., CMII exposed to CeD- or CTL-derived MoDCs. Residual MoDCs were removed from the CGM through centrifugation at 1000× *g* for 30 min. The supernatant representing cell-free CGM was stored at −80 °C for downstream applications. 

#### 4.3.1. Isolation of Exosomes from CGM

Exosomes were pelleted from CGM (10 mls) following ultracentrifugation at 100,000× *g* for 90 min. The exosomal pellet was re-suspended in 50 mL T-PER reagent (Thermo Scientific, Waltham, MA, USA, cat. # 78510) with the addition of protease-inhibitor cocktail (Thermo Scientific, Waltham, MA, USA, cat. # 87785). Protein concentration was determined with a bicinchoninic acid assay (Thermo ScientificWaltham, MA, USA, cat. # 23225).

#### 4.3.2. Western Blotting Detection of Exosomal HLA-DQ and CD63

Proteins (3 μg per lane) mixed with NuPage LDS sample buffer (1X, Thermo Scientific, Waltham, MA, USA cat # NP0008) were separated by electrophoresis in pre-cast 12% Bis Tris NuPage gels (1.0 mm × 12 well, Thermo Scientific, Waltham, MA, USA cat. # NP0342BOX). HLA-DQ and CD63 were identified by Western blotting using antibodies indicated below and goat anti-mouse or human HRP secondary antibodies (1 in 10,000 in TBS-T) and a chemiluminescence detection kit (Thermo Scientific, Waltham, MA, USA, cat # 89880). Images were obtained using a ChemiDoc imager (Bio-Rad, Hertfordshire, United Kingdom). Nytran filters were stained in Ponceau S solution (Thermo Scientific, Waltham, MA, USA, cat # ab270042) to enable normalization using common non-specific banding.

### 4.4. T cell Isolation from Peripheral Whole Blood and Cultivation in CGM

Peripheral whole blood from CTL participants was acquired (10 mL) into collection tubes containing an EDTA anticoagulant. In order to avoid internalization of the CD3 molecule, whole blood was placed on ice. CD3^+^ T cells were isolated by magnetic bead isolation according to the manufacturer’s protocol (Dynabeads^®^ FlowComp™ Human CD3, Thermo-Fisher Scientific, Waltham, MA, USA; cat. number 11365D). In brief, the FlowComp™ Human CD3 Antibody (12.5 μL/mL whole blood) was added directly to anti-coagulated whole blood on ice. After mixing and incubation on ice for 10 min, isolation buffer (20 mL; Ca^2+^ and Mg^2+^ free PBS supplemented with 0.1% BSA and 2 mM EDTA) was added, followed by centrifugation for 15 min at 350× *g*. The supernatant was discarded and the pellet resuspended in FlowComp™ Dynabeads^®^ (375 μL) and mixed well by vortexing. After incubation with rotation for 15 min at RT, isolation buffer (20 mL) was added and placed on a magnet for 3 min. CD3 negative cells were removed from the supernatant. The pellet was resuspended in FlowComp™ Release Buffer (5 mL) containing biotin in 0.1% BSA and 2 mM EDTA, incubated for 10 min RT and placed again on the magnet. CD^+^ T cells were isolated from the supernatant by centrifugation for 8 min at 350× *g*, and resuspended in CGM. Harvested CD3^+^ T cells were phenotyped by flow cytometry and IL-2 cytokine secretion (see below). Verified CD3^+^ T cells (1 × 10^5^/well/24-well dish (TPP, Trasadingen, Switzerland)) were cultivated for 6 days in CGM at 37 °C in 5% CO_2_ atmosphere and 95% relative humidity.

### 4.5. Co-Cultivation of MoDCs and T Cells

Mature MoDCs (1 × 10^5^/well) and CD3^+^ T cells (1 × 10^6^/well) were co-cultivated on poly-L-lysine-treated coverslips in 24-well tissue culture plates (TPP, Trasadingen, Switzerland) at 5% CO_2_/95% relative humidity in CMI for 6 days. During this time, co-cultured cells were washed in 1 × PBS after removal of CMI and serially fixed in 4% paraformaldehyde on days 1, 3 and 6. Coverslips were transferred to microscopic slides and MoDC–T cell interaction determined by proximity evaluation using an Olympus BX53M microscope (Olympus, Tokyo, Japan).

### 4.6. Identification and Quantification of ELMs in HLA-DQA1 and HLA-DQB1

*HLA-DQA1* and *HLA-DQB1* allele transcript sequences derived from the Immuno-Polymorphism Database—International Immunogenetics Project—Human Leukocyte Antigen (IPD-IMGT/HLA) [31,47] were used for determining the presence or absence of previously reported ELMs (i.e., ACCAGCCU; CAGUGAGC; UAAUCCCA) [29,30]. Motifs (ACCAGCCU; CAGUGAGC; UAAUCCCA i.e., ≥5 bp from 3′) were counted per *HLA-DQA1* and *HLA-DQB1* allele transcript. *HLA-DQA1* and *HLA-DQB1* alleles associated with CeD protection or risk [39] were evaluated for full and truncated versions of these ELM sequences.

### 4.7. Immunohistochemical Detection and Confocal Microscopic Visualization of HLA-DQ on T Cells Exposed to CGM

T cells were cultivated in CGM, as outlined above, on glass coverslips. After 6 days, T cells were washed in 1 × PBS twice for 5 min and fixed in 4% paraformaldehyde for 15 min. After another wash in 1 × PBS for 5 min, T cells were permeabilized in 1× TBST (1× TBS, including 0.5% Triton™ X-100 (Sigma-Aldrich, St. Louis, MO, USA)) for 1 h. Following one wash for 5 min in 1× PBS, T cells were placed in blocking solution (1 × TBS containing 5% bovine serum albumin and 0.5% Triton X-100) for 1 h at RT. T cells were then exposed to FITC-conjugated HLA-DQ (clone Tu169 specific for HLA-DQ1 and DQ2 antigens, BD Biosciences, CA, USA; 1:100 in blocking solution; details above) overnight at 4 °C. Samples were washed 3 times for 15 min in 1 × PBS at RT. Coverslips were air-dried in the dark, mounted in ProLong^®^ Gold Antifade reagent (Cell Signalling Technology, Danvers, MA, USA.; cat. # 8961S) containing DAPI and placed on glass microscopic slides. Images were acquired as single XY planes with the pinhole set to Airy1 on an inverted confocal laser scanning microscope Leica TCS SP5 (DMI6000, Leica Microsystems, Mannheim, Germany) using an HCX PL Fluotar lambda blue 40.0 × 0.75 dry objective.

### 4.8. IL-2 ELISA Detection as a Marker of T cell Activation

T cells were cultivated in CGM for 6 days, as indicated above. Subsequently, the CGM was removed, centrifuged at 2000× *g* for 10 min to remove any cells and processed for the cytokine IL-2 using Human IL-2 SimpleStep ELISA^®^ Kit (Abcam, Cambridge, UK; cat # ab270883). This kit used an affinity tag labeled capture antibody and a reporter conjugated detector antibody that immunocaptured the sample IL-2 in solution. The entire complex (capture antibody/analyte/detector antibody) was immobilized via the immuno-affinity of an anti-tag antibody coating the well. Samples or standards were added to the wells, followed by the antibody mix. Following incubation, the wells were washed to remove unbound material. Development solution supplied with the kit was added with incubation enabling HRP catalysis, and the generation of a blue color. The reaction was stopped by the addition of Stop Solution (supplied in the kit) completing any color change from blue to yellow. A signal was generated proportionally to the amount of bound IL-2 and optical density measurements obtained at 450 nm. Duplicate readings were averaged for standards and samples.

### 4.9. Flow Cytometry and FACS Staining

Isolated T cells were cultivated for 6 days in 24-well Tissue Culture Test Plates (TPP, Trasadingen, Switzerland) in the presence of CGM at 37 °C in 5% CO_2_ atmosphere. On day 6, T cells were manually removed using a cell scraper, briefly centrifuged at 400× *g* and resuspended in 100 μL RPMI 1640 medium (Sigma-Aldrich, St. Louis, MO, USA) with monoclonal antibodies (see below) added according to the manufacturer’s recommendation, and incubated for 20 min at 4 °C in the dark. The following monoclonal antibodies were used: Mouse FITC-conjugated anti-Human HLA-DQ (clone Tu169; BD Biosciences, San Jose, CA, USA), PE Mouse Anti-Human CD127 (clone R34.34; Beckman Coulter Company, Brea, CA, USA), PC5 Mouse Anti-Human CD25 (clone B1.49.9; Beckman Coulter Company, Brea, CA, USA) and Anti-Human CD3 PerCP (clone SK7; EXBIO, Prague, Czech Republic), Anti-Human CD63 APC (clone MEM-259; EXBIO, Prague, Czech Republic). An additional wash was performed after incubation. T cells were resuspended in cold PBS (Sigma-Aldrich, St. Louis, MO, USA) with analysis performed on a BD FACSVerse flow cytometer (BD Biosciences, San Jose, CA, USA). Data were analyzed with BD FACSuite™ Software. All images and graphs were generated and assembled using GraphPad PRISM v3.0 and Inkscape v1.0.1 with axes reflective of voltage and fluorescence detector settings.

### 4.10. Statistical Analysis

Data comparison was performed using unpaired Student’s *t*-test in GraphPad PRISM 3.0 or Microsoft Excel software. Variation was represented as standard error. Values of *p* ≤ 0.05 were considered significant.

## Figures and Tables

**Figure 1 ijms-23-06102-f001:**
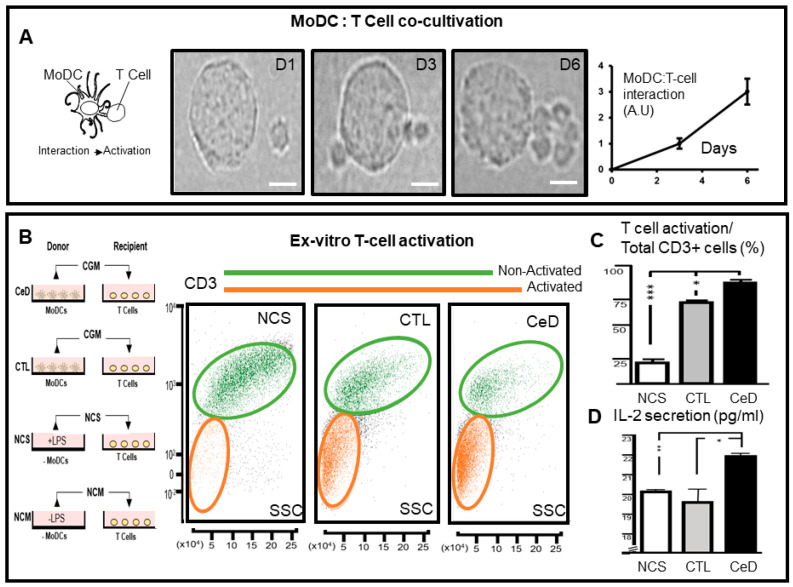
T cell activation upon direct interaction with monocyte-derived dendritic cells (MoDCs) of Celiac disease patients is circumvented by exposure to conditioned growth media. (**A**) Schematic illustration of MoDC interaction with a T cell, which conventionally leads to activation of the latter (left). Representative time-lapse light microscopic images of a MoDC co-cultured with T cells showing no interaction on day 1 (D1) and increasing intercellular interaction between days 3 and 6 (D3; D6, respectively; middle). Scale bar = 5 μm. MoDC: T cell-interaction plot showing increase in intercellular interaction over time (interaction values presented as arbitrary units; *n* = 3; right). (**B**) Illustrative schematic of the experimental overview (left). Monocyte-derived dendritic cells originally sourced from donor Celiac disease patients (CeD) or healthy controls (CTLs), were grown for 6 days in growth medium and transferred (i.e., conditioned growth media—CGM) to T cell recipients isolated from CTLs. Non-conditioned media (NCM) stimulated with LPS (NCS) was also transferred to CTL-T cells. Flow cytometric scatter plot of CD3^+^, a T cell lineage marker versus side scatter (SSC). CD3 is downregulated in recipient T cells exposed to CGM from CTLs and CeD indicative of their activation status (orange line above, orange color gating). T cells exposed to NCS remained non-activated (green line above, green color gating) (middle). (**C**) Histograms of percentage activated T cells per total CD3^+^ T cell population (upper) and (**D**) IL-2 secretion post NCS-, CTL- or CeD-CGM exposure for 6 days (right). * *p* ≤ 0.05, ** *p* ≤ 0.005, *** *p* ≤ 0.001.

**Figure 2 ijms-23-06102-f002:**
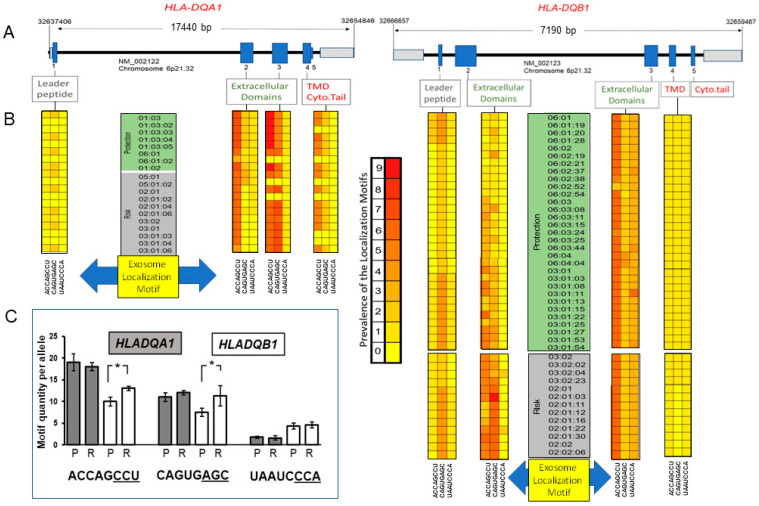
Higher prevalence of exosome-targeting sequences in HLA-DQA1 and HLA-DQB1 alleles associated with Celiac disease risk. (**A**) Schematic of genomic architecture of HLA-DQA1 and HLA-DQB1 located on chromosome 6p21.32. Genomic positioning is provided in black numbering marking the 5′ and 3′ ends (left to right) for each gene. Of note, HLA-DQA1 and HLA-DQB1 are transcribed in opposite directions. The reference sequence source is indicated under the introns (black) and exons (blue) are indicated as well as 5′ and 3′ untranslated regions (UTR) (gray). HLA peptide regions translated from exons are shown. (**B**) Heat map of the prevalence of exosomal localization motifs (ELMs) within Celiac disease HLA-DQA1 (left) and HLA-DQB1 (right) protective (green) or risk (gray)-associated alleles. Exosome localization motif sequence shown below. ELM prevalence indicator (center). Zero number represents absence of ELM (pale yellow); numbers 1–9 represent the numerical frequency of the ELM (dark yellow intensifying to red), respectively. (**C**) Histograms representing the combined ELM quantity per HLA-DQA1 (gray bars) or HLA-DQB1 (white bars) CeD protective (P) or risk (R)-associated alleles. ≥5 bps represented as ELM with underlined bases. * *p* ≤ 0.05.

**Figure 3 ijms-23-06102-f003:**
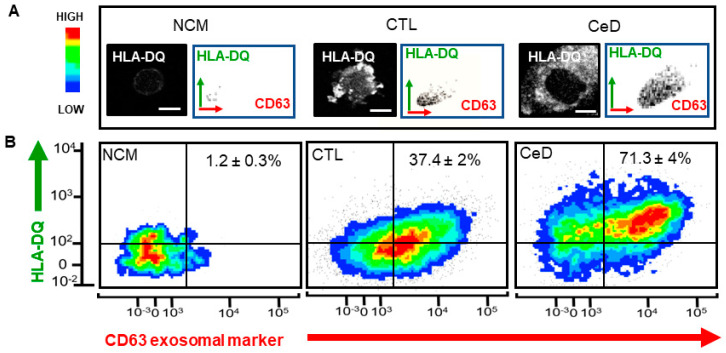
Enhanced exosomal HLA-DQ in T cells exposed to conditioned growth media (CGM) from CeD-sourced MoDCs. (**A**) Representative black and white confocal micrographs of HLA-DQ (DQ; white signal) within a T cell exposed to non-conditioned growth media (NCM) or CGM. Scale bar = 10 µm. Side scatter (SSC) plots of T cell populations from upper-right-quadrant gating (**B**) demarcating predominant fraction with HLA-DQ within CD63-labeled exosomes. Left: Naïve T cells, i.e., exposure to NCM; middle: T cells exposed to CGM from monocyte-derived dendritic cells (MoDCs) from healthy individuals—CTL; right: T cells exposed to CGM from MoDCs from CeD patients—CeD. (**B**) Dot blot density cytograms of FITC conjugated HLA-DQ versus APC-conjugated CD63 providing an indication of signal intensity and extent of CD63 exosomal colocalization. The blue/green/yellow/red/hot spots represent increasing numbers of events (low to high) resulting from discrete cell populations. Percentage cell count per total T cell population data shown.

**Figure 4 ijms-23-06102-f004:**
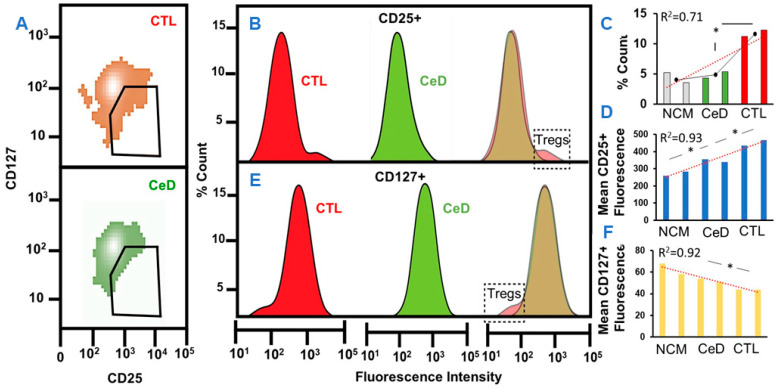
T cells exposed to CGM from CeD elicit compromised induction to Tregs (T—regulatory cells) compared to CTLs. (**A**) Median fluorescence intensity of CD127 versus CD25 in T cells exposed to CGM from CTL or CeD with gating shown. (**B**) Curved flow cytometric histograms (CFCHs) of CD25-labeled T cells showing increased percentage count of Tregs/total T cell population, 6 days after exposure to CGM from healthy CTL-MoDCs (red) compared to CGM from CeD-MoDCs (green). Overlap of CFCHs indicating upregulated CD25 Treg profile within CTL group (dashed outlined box). (**C**,**D**) Histograms of percentage count of Tregs/T cell population (**C**) and mean fluorescence intensity of CD25 T cells exposed to non-conditioned media (NCM), CGM from CTL-MoDCs or CeD-MoDCs (**D**). (**E**) Curved flow cytometric histograms (CFCH) of CD127-labeled T cells showing increased percentage count of Tregs/total T cell population, 6 days after exposure to CGM from healthy CTL-MoDCs (red) compared to CGM from CeD-MoDCs (green). Overlap of CFCHs indicating downregulated CD127 Treg profile within CTL group (dashed outlined box). (**F**) Histograms of mean fluorescence intensity of CD127 T cells exposed to non-conditioned media (NCM), CGM from CTL-MoDCs or CeD-MoDCs. Trendline (red dashed line); mean value per group (black dots). R^2^ provides fitness to the linear curve value. * *p* ≤ 0.05.

## Data Availability

The data presented in this study are available on request from the corresponding author. The data are not publicly available due to ethical restrictions.

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
