# Peer review of "Enhanced Extracellular Transfer of HLA-DQ Activates CD3+ Lymphocytes towards Compromised Treg Induction in Celiac Disease"

_ijms, 2022, doi:10.3390/ijms23116102_

Round 1
Reviewer 1 Report
The manuscript entitled “ Enhanced Extracellular Transfer of HLA-DQ Activates CD3+ Lymphocytes Towards Compromised Treg Induction in Celiac Disease.” by Hudec et al. investigates the impact of soluble factors in conditioned growth medium (CGM) of monocyte derived DC from patients with Celiac disease (CeD) on T cell activity.
The manuscript addresses an important topic and aims to offer greater insight into the pathomechanism of this autoimmune disease. The data provided indicates, that CGM from MoDC derived from CeD patients activates T cells more strongly than CGM from MoDCs of healthy controls. However, the authors provide little to no evidence, that the observed effect of CGM is due to extracellular vesicle (EV)-associated HLA-DQ molecules.
Data regarding extracellular vesicle characteristics is completely missing. Since only crude conditioned growth medium (CGM) was used, it can only be assumed that the transfer of HLA-DQ is extracellular vesicle-mediated. The increased expression of CD63 and HLA-DQ upon exposition to CGM of CeD sourced MoDCs is not sufficient evidence that HLA-DQ was transported by EVs. It is known that HLA-DQ is expressed by T cells upon activation (DOI: 10.1016/j.humimm.2004.01.005 ). Also, CD63, while only minimally expressed on the surface of resting T cells, is present in the cytoplasm and can be found on the surface after activation (DOI:10.4049/jimmunol.173.10.6000 ). Therefore, it cannot be excluded that the observed HLA-DQ and CD63 expression are proteins from the T cells, rather than being transferred from MoDC derived EVs. The confocal microscope images in figure 3 are insufficient to actually identify whether HLA-DQ is in the cell or rather cell membrane-associated.
Thus, Data regarding the characteristics of purified EVs from CGM according to the MISEV guidelines (https://doi.org/10.1080/20013078.2018.1535750) are indispensable. Also, experiments with purified EVs are necessary, if the authors claim EVs as transport vehicle for HLA-DQ. In these experiments the EV numbers should be adjusted, to ensure that the observed differences between CeD and CTL CGM are not due to different vesicle concentrations. Quantification and characterization of the EVs released by CeD and healthy volunteer sourced MoDCs is mandatory. For example, are co-stimulatory proteins (B7 protein) expressed on the EVs? Is HLA-DQ indeed stronger expressed on EVs from CeD MoDCs? The finding that some exon sequences of HLA-DQB1 have an increased prevalence of two exosomal localization motifs described to be found in exosome-enriched RNA does not tell us anything about the targeting of the respective proteins into exosomes.
Comments in more detail:
Figure 1
The characterization of T cells after incubation with CGM is extremely basic. Why weren’t additional activation markers (e.g. HLA-DR, CTLA-4) included? The almost complete lack of CD3 expression after 6 days of incubation with CGM is quite baffling. Did these cells still express CD4 (or CD8)? Can be ruled out that the lack of CD3 detection is not due to blocking of the anti-CD3 antibody binding (by soluble HLA proteins in the CGM).
Figure 3
It would be interesting to be able to compare the staining for HLA-DQ and CD63 without permeabilization. What is depicted in the right box in A? It looks like the same setting as in B, but the cell distribution seems to differ substantially. Please clarify. The figure legend is quite difficult to understand.
Figure 4:
The presentation of FACS data to quantify the Treg population is highly unusual and also inaccurate. To accurately determine regulatory T cells, a gating strategy is necessary: Treg are simultaneously CD127 low AND CD25 high (as activated T cells are also CD25 high). Here, obviously, only one OR the other marker was analyzed, so that Tregs cannot be clearly identified. A dot blot analysis of CD4 (or CD8+) cells would enable to determine the percentage of activated (CD25highCD127+), not activated (CD25neg/low, CD127+) and regulatory T cells (CD25high, CD127low/neg). The additional analysis of FOXP3 as a Treg marker would facilitate the precise identification of Treg cells. In the diagrams B - C, the difference between the two bars of the same color remains unclear. Do they represent two different donors?
Page 10, line 5: the stated fold increase of 0.9 can’t be right. Should probably state: 1.9-fold increase..?!
Reviewer 2 Report
General comments
The conventional cell-mediated antigen presentation by antigen-presenting cells has evolved into antigen presentation by exosomes derived from antigen-presenting cells such as MO-dendritic cells. This investigation raises concern about the release of exosomes by MO-derived DCs for Celiac disease based on the response of T-regulatory cells to exosomes in the context of tolerance loss to gluten, the peptides of which are the causal factor for Celiac Disease.
This study reports that Conditioned Growth Medium sourced from CeD patients in contrast to that of the control Medium, impacting T-cells from healthy individuals, as evidenced by augmented IL-2 production and alteration in T-regs characteristics. T-regs are known to play a role in tolerance against food allergens. Los of their characteristics is envisaged as a cause for CeD pathogenesis.
This study also identifies the increased presence of HLA-DQ within exosomes stemming from T-cells treated with CeD–CGM, in contrast to those exposed to a normal growth medium.
In essence, this study offers novel insight into the basis of immune dysregulation associated with CeD.
Specific comments:
Although this manuscript is emanating from experienced investigators like Drs. O’Leary and Dr. Jurickova, it appears that this manuscript is written by novice writers. Indeed, this is an interesting manuscript. But it suffers due to poor writing in different sections in different ways. Both senior authors and junior authors should read this manuscript from a reader’s perspective to understand the complexities of this manuscript. Revision requires good training of the mind.
Major comments
- As it is well-known CD63 is also a common component of the lysosomal membranes, Microvesicles (MVs not exosomes) and apoptotic bodies, it would have been better if the author tested CD9 and CD81, the other components of exosomes, not necessarily found on MVs and apoptotic bodies.
- Immunochemical detection and visualization of HLA-DQ are totally not convincing. They should have used a monoclonal antibody for HLA-DQA or DQB or both. FITC conjugate HLA-DQ, is it DQA or DQB? This is totally misleading and unconvincing. Native DQA and DQB should be documented on T cells immunochemically.
- Or at least a polyreactive anti-HLA-DQ monoclonal antibody should have been used to document the HLA-II on the exosomes.
- It would have been a classical work if the exosomes isolated from CeD patients’ Mo-DCs with magnetic beads were tested for the presence of HLA-DQA/DQB/CD63/CD9/CD81.
Minor comments
- The Abbreviations hamper continuous reading of the manuscript and create headaches:
- CTLs for immunology reader means cytotoxic T lymphocytes: however the author uses the abbreviation CTL for the first time on page 4. Did they say what C, T, and L stand for?
- They used CTL 38 times and nowhere could I find the elaborated version, although I could assume what it could be.
- An evidence for non-sensical abbreviation is presented in the following sentence in pages 6-7. “Non-conditioned growth media (GM) with stimulatory liposaccharide (NCS) or without LPS (Non-conditioned media (NCM))”. Is GM refers to Conditioned media or growth media?; Does NCS elaborates to stimulatory liposaccharide? Is Non-condition growth media not Non-conditioned media (NCM)?
- ELM is used in 19 places in the text. First in Title 2.2 on page 7. What does ELM stand for? I went through 8 and 9, on page 9 they clarified what ELM is for as exosomal localization motifs. Again a sentence after that they write Exosomal localization motif without ELM!!! Again on page 20, they state ELM with elaboration! Is this way to write a manuscript?
- CMI is first introduced on page 5. what is its elaboration? In Section 4.3 it is stated ” CMI (CM with the inclusion of human granulocyte[1]macrophage colony‐stimulating factor (GM ‐ CSF)”
- Then comes CMII on page 9 end. CMII jumps to page 18 where it is stated “CMI was replaced with CMII (ie. CMI with instead exosome depleted foetal bovine serum..”
- In page 2 para 1
The MHC-TCR bridge binds the MoDC to the T lymphocyte- is it transliteration of another language?
- Page 19
Generated MoDCs (CD11c+ , CD14‐ ) were divided into 6‐well plates with the addition of lipopolysaccharide (LPS) (100 ng/mL; Sigma ‐ Aldrich, MO, USA) and cultivated for 24 hrs (Schoeller Pharma, Prague, Czech Republic) to induce MoDC maturation.
What does (Schoeller Pharma, Prague, Czech Republic) stand for? LPS? Or for what?
Recommendations:
This manuscript contains a novel hypothesis but is not satisfactorily verified. This partial verification can be considered for publication, provided the authors undertake to revise extensively the manuscript. If comments major comments cannot be carried out, they should at least be explained and discussed from a future perspective.
Above all,
- the Abbreviations should either be minimized or deleted or a table for abbreviations should be provided in the manuscript
- Must be edited for Scientific and English writing
The manuscript needs revision before considered for publication.
